# Evaluation of Genetic Testing in a Cohort of Diverse Pediatric Patients in the United States with Congenital Cataracts

**DOI:** 10.3390/genes14030608

**Published:** 2023-02-28

**Authors:** Jennifer L. Rossen, Brenda L. Bohnsack, Kevin X. Zhang, Alexander Ing, Andy Drackley, Valerie Castelluccio, Hanta Ralay-Ranaivo

**Affiliations:** 1Department of Ophthalmology, Northwestern University Feinberg School of Medicine, Chicago, IL 60611, USA; 2Division of Ophthalmology, Department of Surgery, Ann & Robert H. Lurie Children’s Hospital of Chicago, Chicago, IL 60611, USA

**Keywords:** congenital cataract, diagnostic yield, variant reclassification

## Abstract

The aim of this study was to evaluate the diagnostic yield from prior genetic testing in a 20-year cohort of pediatric patients with congenital cataracts. A retrospective review of patients with congenital cataracts who underwent genetic testing was completed from 2003–2022. The diagnostic yield of the test was determined by variant classification and inheritance pattern. Variants from initial testing underwent reclassification in accordance with ACMG-AMP (American College of Medical Genetics and Genomics—American Association of Molecular Pathology) 2015 or 2020 ACMG CNV guidelines. A total of 95 variants were identified in 52 patients with congenital cataracts (42 bilateral, 10 unilateral); 42 % were White, 37% were Hispanic, 8% were Black, and 6% were Asian. The majority of patients (92%) did not have a family history of congenital cataracts but did have systemic illnesses (77%). Whole exome sequencing and targeted congenital cataract panels showed diagnostic yields of 46.2% and 37.5%, respectively. Microarray had the lowest yield at 11%. Compared to the initial classification, 16% (15 of 92 variants) had discrepant reclassifications. More testing is needed, and an increased focus is warranted in the field of ocular genetics on congenital cataracts, particularly in those with systemic illnesses and no family history, to advance our knowledge of this potentially blinding condition.

## 1. Introduction

Legal blindness affects 0.4 per 1000 children in developed countries and 1.2 per 1000 children in underdeveloped countries, accounting for approximately 1.4 million children worldwide [1]. Inherited diseases have been estimated to cause 23–44% of childhood vision loss, but there is no national registry of etiologies for vision loss in the United States to establish a firm prevalence [2]. While genetic tests in children with ocular diseases can provide information on diagnosis, prognosis, treatment options, and family planning, tests have traditionally been used sparsely due to cost and not being covered by insurance. Modern, often free, targeted genetic panels using next-generation sequencing platforms for ocular diseases, such as retinal dystrophies and congenital cataracts, have recently increased the frequency of genetic testing. However, recent research efforts have prioritized retinal dystrophies due to the advent of novel gene therapy for such disorders [3]. The same attention has not been paid to congenital cataracts over the last 5 years, with a down-trend in the frequency of publications on congenital cataracts since its height in 2016 [4]. 

Congenital cataracts merit a similar degree of scientific attention as retinal dystrophies. Their prevalence is equivalent to congenital cataracts which affect 1.7–3 in 10,000 children younger than 15 years old, while retinal dystrophies affect 2.2 in 10,000 children under 16 years old [1,5,6]. Furthermore, childhood-onset cataracts are often associated with systemic illnesses, and early genetic testing early can identify potentially treatable conditions such as galactosemia and cerebrotendinous xanthomatosis [7,8]. 

The management of congenital cataracts is more challenging than adult-onset cataracts. Recent advancements in adult cataract surgery are not applicable to most children. Post-operative care is challenging and frequently associated with late-onset complications [9]. Genetic variants associated with congenital cataracts have been researched as targets for potential novel therapies, but further research is warranted to find better treatments [7]. As a first step in better understanding the pathogenesis of congenital cataracts and ultimately discovering new treatments, this study was developed to evaluate the genetic diagnostic yield from prior genetic testing in a cohort of diverse pediatric patients over the past 20 years. 

## 2. Materials and Methods

### 2.1. Clinical Assessment

A retrospective case series identified all patients with congenital cataracts who presented to the Ann & Robert H. Lurie Children’s Hospital of Chicago between January 2003 and July 2022. Inclusion criteria included available results from prior genetic testing. Patients with retinal dystrophies or hereditary vitreopathy (which can have associated cataracts), a history of inflammation, prior steroid use, or trauma were excluded. Clinical information collected included age at presentation, race/ethnicity, family history of cataracts, additional ocular diagnoses, ocular findings (including characteristics of the cataract and other ocular abnormalities), systemic comorbidities, and ocular surgical history. This study protocol adhered to the tenets of the Declaration of Helsinki and received approval from the Lurie Children’s Hospital Institutional Review Board (IRB 2021-4110).

### 2.2. Molecular Investigations

Varying genetic tests were completed in this patient cohort, including microarrays, targeted next-generation sequencing, confirmatory/variant assays, whole exome sequencing (WES), and mitochondrial sequencing. The decision on the type of test was predominantly determined by the genetic specialist after discussion with patients and parents on testing options and associated costs. Patients received genetic counseling by a genetic counselor or geneticist associated with their testing as per the standard of care. Many of the biological parents of patients also underwent genetic testing for the evaluation of inheritance patterns and family planning. 

### 2.3. Analysis

The results were divided into different arms based on the type of testing: microarray, targeted testing for ocular disease, expanded testing (mostly WES), and testing done for other systemic reasons. Classification of variants identified upon testing prior to 2015, when Richards et al. proposed guidelines for the interpretation of variants, were evaluated and recorded according to a 3-tier system (pathogenic/likely pathogenic (P/LP), variant of uncertain significance (VUS), and benign/likely benign (B/LB)); after 2015 guidelines were implemented, variants were classified according to the standardized 5-tier system (P, LP, VUS, B, LB) [10]. Enriched variants were cross-referenced against an online database of genes associated with inherited or age-related cataracts (CatMap, https://cat-map.wustl.edu/, accessed on 28 April 2022) [11]. Commercially available panels were also cross-referenced (Appendix A). For genes not identified in CatMap or testing panels, the literature was reviewed for prior reports of an association between genes with identified variants and cataracts. Age at the time of surgery, which was an indicator of cataract severity, was divided into 4 categories: less than 1 year of age (most severe), 1 to 10 years of age (moderate severity), >10 years of age, or has not yet required surgery (less severe). The age 1–10 years was chosen for moderate severity since deprivational amblyopia was possible in this age group.

For pathway enrichment analysis of variant genes, gene ontology (GO) analysis was performed using the PANTHER online database of gene annotations and classifications [12,13]. A false-discovery rate corrected P-value of 0.05 was selected as the cutoff criterion. 

Because the genetic variants identified in our cohort were identified over a considerable time span, variant reinterpretation, conducted by 2 genetic counselors (AI, AD) who did not have knowledge of the initial classifications at the time of reinterpretation, was performed to obtain the variants’ most up-to-date determinations of clinical significance. Variants were re-interpreted in accordance with ACMG-AMP (American College of Medical Genetics and Genomics – American Association of Molecular Pathology) 2015 sequence [10] or 2020 ACMG-ClinGen CNV guidelines [14]. These reinterpretations included a review of publicly available databases (e.g., ClinVar [15], Leiden Open Variation Database [16]), and the latest biomedical literature for data relevant to each variant. Variant minor allele frequency and gene constraint data were obtained from the Genome Aggregation Database [17] (gnomAD) v2.1.1 and v3.1.2. In silico pathogenicity predictions were performed using REVEL [18] (benign: ≤0.290; pathogenic: ≥0.644 [19]) and the splicing prediction algorithms SpliceSiteFinder-like [20], MaxEntScan [21], NNSplice [22], GeneSplicer [23] (predicted impact: ∆ ≥ 10% between wild-type and variant), and SpliceAI [24] (predicted impact: ≥0.2; no impact: <0.2). Updated variant classifications were then compared to the original classifications.

## 3. Results

A total of 52 patients were identified with congenital cataracts (42 bilateral) that underwent genetic testing (Appendix A), with demographics summarized in Table 1. In total, 42% percent were White, 37% Hispanic, 8% Black, 6% Asian, and 8% other; 77% of patients had concordant systemic illnesses, and only 8% had a family history of congenital cataracts. In addition, 58% of patients underwent lensectomy (83% bilaterally); 53% of patients who had surgery were infants less than 1 year of age at the time of lensectomy, 27% were between 1–10 years old, and 20% were older than 10 years old. 

### 3.1. Microarray Results

Eighteen patients underwent microarray testing as their primary genetic modality from 2006 to 2021 (Appendix A and Table 2). All patients had other systemic conditions, and 72% (13 patients) had other ocular anomalies. Only 2 of the 18 patients (11%) had a positive genetic diagnostic yield from microarray testing. Patient #3 had a deletion in chromosome 21, including the *RUNX1*, *DYRK1A*, and *KCNJ6* genes, classified as likely pathogenic as variants in *DYRK1A* have been associated with bilateral congenital cataracts [25]. Patient #1 had a deletion on chromosome 11, including *PAX6*, which is a gene well-known to be associated with cataracts and other types of anterior segment dysgenesis. Patient #5, with bilateral cortical cataracts, many systemic illnesses, and left congenital ptosis, had a large gain and loss in 7q, including several genes; however, none of these genes have known associations with congenital cataracts. Three patients had a unilateral cataract, all of which had negative diagnostic yield.

### 3.2. Targeted Genetic Testing for Ocular Phenotypes

Thirteen patients obtained targeted genetic testing for congenital cataracts (*n* = 8) or other anterior segment dysgenesis (*n* = 5) (Appendix A and Table 3). Only one patient in this cohort had a family history of congenital cataracts, and one patient had other serious systemic illnesses/syndromic conditions. Six patients (46.2%) had a positive diagnostic yield, of which three were positive with the congenital cataract panels (Table 3). The three patients with the positive diagnostic yield on the cataract panels had severe cataracts at birth, requiring surgical removal in infancy (<1 year old). One of the three patients with a unilateral cataract in this cohort had a positive diagnostic yield (33.3%). 

### 3.3. Whole Exome Sequencing/Expanded Testing

Thirteen patients underwent whole exome sequencing (WES), often also with testing of the mitochondrial genome, and one patient underwent only testing of the mitochondrial genome (Appendix A and Table 4). All patients had systemic illnesses/syndromic conditions, and only one patient had a family history of congenital cataracts. Mitochondrial genome testing of patient #45 was not found to have any variants that could explain the phenotype. Six out of thirteen patients who pursued WES (46.2%) had a positive diagnostic yield. Three patients (23%) had inconclusive results. When comparing diagnostic yield to the age at time of surgery, two of the three patients (67%) who required surgery during infancy (most severe) had positive diagnostic yield. Three patients had a unilateral cataract, one had a positive diagnostic yield (33.3%), and one (patient #43) had inconclusive results. 

### 3.4. Genetic Testing for Other Systemic Phenotypes

Seven patients with congenital cataracts and other systemic diseases obtained targeted testing for a separate diagnosis (e.g., epilepsy panel, autism panel, familial variant/confirmatory research testing) (Appendix A and Table 5). Only one patient (#46) had a positive diagnostic yield, and three others had inconclusive results (Table 5). Patient #46 underwent a congenital muscular dystrophy panel and was found to have a pathogenic variant in *POMT1*, a gene that has been associated with congenital cataracts [26]. 

### 3.5. Functional Enrichment Analysis of Variant Genes

All genes with variants identified in patients in this study were collated, and functional enrichment analysis was performed to further study potential pathways in three major categories (biological process, cellular component, and molecular function). Significantly enriched terms (*p* < 0.05) with their corresponding gene ratios are displayed in Figure 1. A total of 58 genes were used in the analysis, retrieving 22 distinct gene ontology terms.

### 3.6. Variant Reclassification

A total of 92 sequence and copy number variants were reinterpreted from 52 patients with congenital cataracts. Compared to the initial classifications, 16% (15/92) had discrepant classifications following reinterpretation, 53% (8/15) represented changes to the degree of certainty (e.g., Pathogenic to Likely Pathogenic), and 46% represented changes to the classification tier, with almost all of these variants reclassifying from VUS to Likely Benign or Benign. None of these updated classifications would have definitively transformed a previously inconclusive report into a diagnostic one; however, one variant in a gene associated with both autosomal dominant and autosomal recessive inheritance was reclassified from VUS to LP, and reclassifications would have resulted in negative reports instead for 6% (1/15) of individuals with previously inconclusive results. 

## 4. Discussion

### 4.1. Diagnostic Yield

Identification of genetic causes of congenital cataracts has lagged behind other ocular diseases, such as retinal dystrophies, due to limited accessibility of testing. Thus, information regarding gene variants is sparse. While there are several commercially available targeted panel tests for congenital cataracts (Appendix A), the first step in improving overall genetic evaluation is reviewing the diagnostic yield of prior genetic testing results. 

A major challenge is the non-uniformity of genetic testing which, depending on clinical presentation, family history, systemic findings, and insurance coverage, includes whole exome sequencing, targeted gene panels, and microarray. Each of these tests shows different diagnostic yields, which reflects their specific limitations and our limited understanding of the genetics of congenital cataracts in the United States (USA). 

In the current study, whole exome sequencing showed the highest diagnostic yield for congenital cataracts at 46%. This is similar to a report by Reis et al. which found a diagnostic yield of 39% using whole exome sequencing [27]. Causative pathogenic variants were identified in 9 of 23 pedigrees (19 White); however, in contrast to our study in which only a small percentage (8%) of patients had a family history of cataracts, all of the pedigrees in Reis et al. had autosomal dominant congenital, juvenile, or early onset cataracts in multiple family members across at least 2 and up to 4 generations [27]. Thus, it is interesting that their diagnostic yield was lower than ours, but this may be due to a difference in 9 years between studies and the increased number of genes and variants that have been identified in the interim. The advantage of whole exome sequencing is that it aids in the identification of new genetic causes of congenital cataracts; however, this test can be time-consuming and expensive. Further, whole genome sequencing, which includes intronic and non-coding inter-gene regions, while even more complex in terms of interpretation and analysis, has shown a greater ability to identify genetic causes of congenital cataracts within these key regulatory regions [28]. 

Targeted gene panels have shown a wide range of diagnostic yields. A group in the United Kingdom and a second in Australia showed much higher diagnostic yields compared to ours of 38%. In Gillespie et al., the diagnostic yield of a 115-gene targeted panel applied to 36 patients in the United Kingdom was 63% for syndromic congenital cataracts and 85% for isolated congenital cataracts, for an overall rate of 75% [29]. Ma et al. used a 55-gene-targeted panel in 52 Australian patients and found a 67% diagnostic yield [30]. In contrast, Sun et al. showed a diagnostic yield of 40% using targeted sequencing of 12 genes in 25 Chinese families [31]. The discrepancy between these studies and our lower diagnostic yield of 38% with available targeted panels may be related to a number of factors. For example, the accessibility of genetic testing is much greater in the UK and Australia with government-funded healthcare. In the U.S., genetic testing, especially for diagnoses that are believed to be less likely to change clinical management, is often not covered by insurance and is expensive. As a result, the genetic results in our study reflect only a small proportion of patients who presented to our institution with congenital cataracts. More recently, industry-sponsored genetic testing with a panel of 66 genes has become available at no cost to patients with bilateral congenital cataracts who are over 18 months of age. With this resource to aid in testing a greater proportion of new patients with congenital cataracts, the diagnostic yield in our population is likely to increase. The number of genes screened in any given panel also strongly influences the diagnostic yield. The greater the number of genes, the higher the probability of a positive result. This is highlighted by the finding that the UK study, which tested for the greatest number of genes (115), had the greatest diagnostic yield compared to the Australian and Chinese panels, which had 55 and 12 genes, respectively [29,30,31]. In our study, different commercially available panels were utilized, which ranged from 66 to 171 genes associated with congenital cataracts. While there are overlapping genes between the various panels, there are differences that also may reflect the overall lower diagnostic yield with this technique. Larger panel sizes and more uniformity of testing may improve diagnostic yields. 

It is also important to note the genetic heterogeneity of the patient populations and the effect on the identification of gene variants. While the UK study did not include race or ethnicity data, the Australian and Chinese cohorts were predominantly White and Asian, respectively. In contrast, our patient population was much more heterogenous, with 42% of patients White and only 6% Asian. To date, there has been a paucity of congenital cataract variant data regarding non-White and non-Asian populations, highlighting the importance of acquiring data from diverse cohorts to aid in variant interpretation, thus increasing the utility of genetic testing in all populations. 

### 4.2. Inconclusive/Negative Results

Inconclusive results are common with genetic testing. A number of patients in our cohort were found to have mutations in genes, which may be causative; however, the entire clinical picture was not consistent with the genetic diagnoses. Patient #40 had a history of spinocerebellar ataxia and moderately sized posterior polar cataracts; WES identified two P variants in gene *SYNE1*, which has been previously reported in congenital muscular dystrophy, a condition known to be associated with cataracts [26]. Patient #43 had two VUSs in *WDPCP* identified by WES; variants in this gene have been reported in association with Bardet--Biedl syndrome, which can present with cataracts along with retinal issues [32]. Patient #44 was found to have a 13q12.11 duplication on WES; *GJA3* resides in this region and is listed in CatMap and included on multiple congenital cataract panels [11]. Analyzing this information is important for expanding genotype-phenotype correlations.

Further, some patients were classified as negative, yet mutations in genes that had not previously been associated with cataracts were identified. Even though patient #50 had a confirmatory research test of an LP variant in the *GRB10* gene, yet no variants in this gene have been previously associated with cataracts in the published literature. Therefore, a negative diagnostic yield for congenital cataracts was reported. Likewise, patient #48 completed a targeted autism panel, which showed LP variants in *KCNB1* and *CSNK2B*, but neither are in CatMap, on commercial cataract panels or previously reported to be associated with cataracts. Patient #47 underwent a cholestasis panel and was found to have a VUS in *NOTCH2*, which has been demonstrated to participate in lens development [33]. Patient #49 possesses a P variant in *NR4A2* and a VUS in *DYNC1H1*. Variants in *DYNC1H1* have been associated previously with autosomal dominant congenital cataracts [11]. Patient #52, with a unilateral cataract, had Hereditary Paraganglioma-Pheochromocytoma Syndrome; he tested positive for the same *SDHB* variant as his affected father (who had the same syndrome and childhood cataracts), but this gene has not been directly implicated in congenital cataracts. Although genetic testing of these patients was considered negative, reporting these results increases the knowledge and potential list for future reference of genes associated with congenital cataracts.

### 4.3. GO Enrichment Analysis

GO enrichment analysis retrieved several expected terms related to embryonic eye morphogenesis (GO:0048048), lens development in the camera-type eye (GO:0002088), and structural constituent of the eye lens (GO:0005212). Protein O-linked glycosylation (GO:1904100) was particularly well represented in our variant gene list, whereby implicated genes *POMT1* and *POMT2* share strong associations with a variety of developmental ophthalmic manifestations, including cataract [34]. Playing critical roles in the eukaryotic protein O-mannosyltransferase (POMT) complex, it is understood that pathogenic variants in *POMT1* and *POMT2* disrupt specific posttranslational protein modifications that contribute to cataractogenesis during early development [35]. In fact, Patient #46 presented with a pathogenic variant in *POMT2* and had stigmata of Walker--Warburg syndrome, including bilateral congenital cataracts, epileptic encephalopathy, and muscle weakness. The utility of functional pathway analysis beyond validating previously documented genotype-phenotype associations lie in the discovery of other genes or gene-regulatory elements that share a common pathogenesis but, by themselves, may not be readily identified in common screening tests.

### 4.4. Variant Reclassification 

The interpretation of genetic variants is highly subject to multiple factors, including date of interpretation, supporting data availability, and usage and application of the ACMG classification criteria. Of the variants that received a new classification upon reinterpretation, a key factor is the presence of the variants in control databases (e.g., gnomAD), which allowed for several variants to be downgraded to Likely Benign or Benign. This is consistent with our initial hypothesis that variants in our study cohort that were reported earlier would benefit from reanalysis as large control databases such as gnomAD were not available to use in the interpretation. However, many inconclusive classifications in our cohort were unchanged after reinterpretation, clarifying that additional data is still required to reach a definitive classification.

Of note, some variants presented a particular challenge in interpretation as the data did not fit neatly in the standardized variant interpretation framework. For example, patient #41 was identified to have the stop-gain variant NM_181523.3(PIK3R1):c.18C > G (p.Tyr6*). Pathogenic variants in *PIK3R1* are associated with the autosomal dominant SHORT syndrome, characterized by Short stature, Hyperextensibility, Hernia, Ocular depression, Rieger anomaly, and Teething delay. However, our literature review did not identify any true loss-of-function (LOF) variants reported in association with SHORT syndrome; all null variants seemingly resulted in protein truncation rather than complete loss of protein expression, which is likely the reason for the initial classification of VUS. During reanalysis, LOF variants were identified in association with another disease in this gene (agammaglobulinemia) but with an autosomal recessive inheritance pattern, suggesting that LOF does indeed cause disease. Per Abou Tayoun et al., evidence criterion PVS1 (null variant in a gene where the loss of function is a known mechanism of disease) should be applied if the variant is predicted to undergo nonsense-mediated mRNA decay and the exon is present in a biologically relevant transcript [36]. For this variant in *PIK3R1*, both conditions were met. Combined with criterion PM2 (absent from controls), this resulted in a classification of Likely Pathogenic. Thus, in addition to a general need for more data overall, this example highlights the need for further gene- and disease-specific curation and knowledge to perform comprehensive variant classification.

### 4.5. Future Research Aims

Overall diagnostic rates of previous testing methods were low in a diverse U.S.-based population of pediatric patients with a high rate of systemic illnesses and a low rate of family history of childhood cataracts. Furthermore, the reclassification of variants from previous testing did not improve the diagnostic yield. Therefore, these patients require additional testing to increase the likelihood of finding a molecular diagnosis. Our future research will aim to provide expanded genetic testing with whole genome sequencing (WGS) for all patients with isolated and syndrome-associated congenital cataracts, with or without a family history of congenital cataracts, that have previously non-diagnostic test results. WGS in a large patient cohort will increase our understanding of genotype-phenotype correlations and elucidate the utility of WGS for congenital cataracts in a U.S.-based population. Furthermore, this information will aid in the identification of new causative genes and variants. By a reverse genetic approach, new causative genes and variants of congenital cataracts identified will provide insight into the pathogenesis of congenital cataracts. The new genes and variants identified will be targets for in vivo zebrafish animal models to better understand lens development and ultimately identify targets for novel therapies. 

## Figures and Tables

**Figure 1 genes-14-00608-f001:**
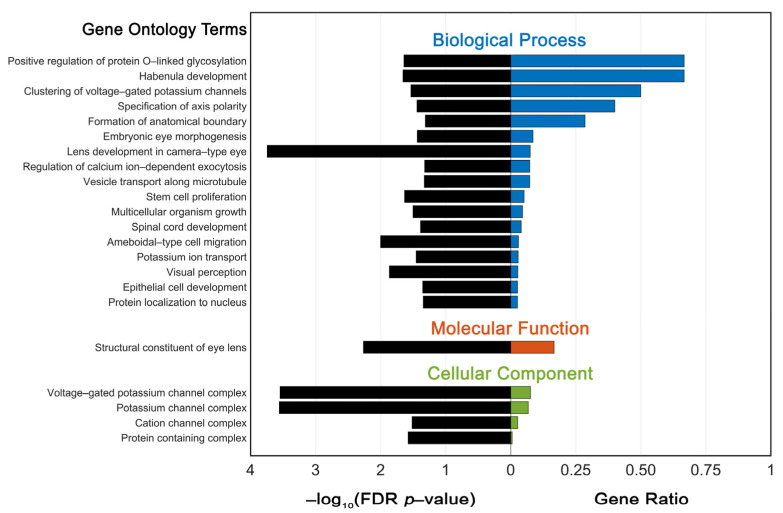
Pathway enrichment analysis depicting significantly enriched gene ontology (GO) terms within categories: biological process, molecular function, and cellular component, as identified from the PANTHER classification database. Fifty-eight total genes, each associated with an identified genetic variant in at least one patient, served as the input. Adjusted P-values were negative 10-base log transformed. FDR = false discovery rate.

**Table 1 genes-14-00608-t001:** Patient Demographics.

	N (%)
**Race/Ethnicity**	
White	22 (42%)
Hispanic	19 (37%)
Black	4 (8%)
Asian	3 (6%)
Other	4 (8%)
**Systemic Illness**	
Yes	40 (77%)
No	12 (23%)
**Family History**	
Yes	4 (8%)
No	48 (92%)
**Lensectomy**	
No	22 (42%)
Yes	30 (58%)
Unilateral	5 (17%)
Bilateral	25 (83%)
<1 year-old	16 (53%)
1–10 years-old	8 (27%)
>10 years-old	6 (20%)

Unilateral = one eye received cataract surgery/lensectomy (percentage are percentage).

**Table 2 genes-14-00608-t002:** Microarray Results.

Patient #	Date of Test	Microarray Findings	Diagnostic Yield Positive
1	2011	11p14.3p11.2(25,958,860–43,778,471)×1	Yes
2	2013	7q31.1(110,950,859–111,234,120)×1	No
3	2020	21q22.12q22.13(35,903,896–39,545,357)×1	Yes
4	2021	2p25.3(1,741,827–1,842,843)×3	No
5	2016	7q35(146,051,998–147,171,974)×3 7q35q36.3(147,171,974–159,138,663)×1	No
6	2010	22q11.22(20,640,000–20,905,000)×1	No
7	2019	Normal	No
8	2009	Normal	No
9	2011	Normal	No
10	2016	Normal	No
11	2018	Normal	No
12	2014	Normal	No
13	2010	Normal	No
14	2008	Normal	No
15	2020	Normal	No
16	2012	Normal	No
17	2005	Normal	No
18	2011	Normal	No

Patient # is the study number provided to each patient to keep confidential protected health information. “Yes” for positive diagnostic yield indicates that the patient had genetic testing results that are believed to explain the patient’s congenital cataracts based on the variant classification and inheritance pattern. Genomic coordinates are provided in human genome reference build 37 (GRCh37/hg19).

**Table 3 genes-14-00608-t003:** Targeted Ocular Genetic Testing Results.

Patient #	Test (Laboratory)	Date of Test	Gene (Variant Classification)	Diagnostic Yield Positive
19	Comprehensive Cataracts Panel (Prevention Genetics)	2020	*CAPN15* (VUS), *CYP27A1* (VUS), *FYCO1* (VUS)	No
20	Early-Onset Bilateral Cataracts Panel (Prevention Genetics)	2021	*CRYBB2* (VUS)	No
21	Comprehensive Cataracts Panel (Prevention Genetics)	2021	*INPP5B* (VUS), *LTBP2* (VUS), *POMT2* (VUS)	No
22	Microphthalmia, Anophthalmia, and Anterior Segment Dysgenesis Gene Panel (Blueprint Genetics)	2019	*PAX6* (LP)	Yes
23	Comprehensive Cataracts Panel (Prevention Genetics)	2019	*NHS* (P)	Yes
24	*PAX6* Gene/Aniridia/Developmental Eye Disorders (GeneDx)	2019	11p13 deletion (P)	Yes
25	Comprehensive Cataract Panel (Prevention Genetics)	2021	*RAB3GAP1* (LP), *RAB3GAP1* (VUS), *BFSP1* (VUS), *BFSP1* (VUS), *ADAMTS10* (VUS), *AGPS* (VUS), *ERCC1* (VUS)	Yes
26	Early-Onset Bilateral Cataracts Panel (Prevention Genetics)	2021	*COL18A1* (VUS), *GALK1* (VUS)	No
27	*PAX6* Sequencing (Emory Genetics Laboratory)	2015	*PAX6* (P)	Yes
28	Microphthalmia, Anophthalmia, and Anterior Segment Dysgenesis Gene Panel (Blueprint Genetics)	2020	No P, LP or VUS	No
29	:Custom sequencing panel with CNV detection for Coloboma Genes (Prevention Genetics)	2018	No P, LP or VUS	No
30	Cataract Panel (Invitae)	2022	*CRYAA* (P), *CRYBB1* ((VUS)	Yes
31	Early-Onset Bilateral Cataracts Panel (Prevention Genetics)	2022	*ERCC2* (VUS), *ERCC2* (VUS)	No

“Yes” for positive diagnostic yield indicates that the patient had genetic testing results that are believed to explain the patient’s congenital cataracts based on the variant classification and inheritance pattern.

**Table 4 genes-14-00608-t004:** Whole Exome Sequencing With or Without Mitochondrial Analysis Results.

Patient #	Test (Laboratory)	Date of Test	Gene (Variant Classification)	Diagnostic Yield Positive
32	XomeDx Plus [WES + mitochondrial genome analyses] (GeneDx)	2019	*PHACTR4* (VUS)	No
33	XomeDx Plus [WES + mitochondrial genome analyses] (GeneDx)	2016	*ALDH18A1* (LP)	Yes
34	XomeDx Plus [WES + mitochondrial genome analyses] (GeneDx)	2017	*DYNC1H1* (LP), *LRP5* (VUS)	Yes
35	XomeDx Plus [WES + mitochondrial genome analyses] (GeneDx)	2020	*KIF1A* (P)	Yes
36	XomeDx [WES] (GeneDx)	2014	*BCOR* (LP/P), *ATP2A3* (VUS)	Yes
37	XomeDx [WES] (GeneDx)	2014	*BCOR* (LP.P)	Yes
38	XomeDx Reanalysis (GeneDx)	2021	No P, LP or VUS	No
39	XomeDx Plus [WES + mitochondrial genome analyses] (GeneDx)	2019	No P, LP or VUS	No
40	XomeDx Plus [WES + mitochondrial genome analyses] (GeneDx)	2017	*SYNE1* (P), *AAAS* (P)	Inconclusive
41	XomeDx Plus [WES + mitochondrial genome analyses] (GeneDx)	2018	*PIK3R1* (VUS)	No
42	Cerebral Palsy Spectrum Disorders Panel (Invitae) & XomeDx Plus [WES + mitochondrial genome analyses] (GeneDx)	2021	*RARB* (LP)	Yes
43	XomeDx Plus [WES + mitochondrial genome analyses] (GeneDx)	2021	*WDPCP* (VUS), *WDPCP* (VUS)	Inconclusive
44	XomeDx [WES] (GeneDx)	2022	13q12.11 duplication which includes *GJA3* (VUS)	Inconclusive

“Yes” for positive diagnostic yield indicates that the patient had genetic testing results that are believed to explain the patient’s congenital cataracts based on the variant classification and inheritance pattern.

**Table 5 genes-14-00608-t005:** Other Targeted Genetic Testing Results.

Patient #	Test (Laboratory)	Date of Test	Gene (Variant Classification)	Diagnostic Yield Positive
46	Dystroglycan-Related Congenital Muscular Dystrophy Panel (Prevention Genetics)	2019	*POMT1* (P)	Yes
47	Cholestasis Panel (EGL Genetics)	2017	*NOTCH2* (VUS)	Inconclusive
48	Autism/ID Xpanded Panel (GeneDx)	2019	*CSNK2B* (LP), *KCNB1* (LP)	No
49	Comprehensive Epilepsy Panel (Lurie Molecular Diagnostic Laboratory) & ExomeNext [WES] (Ambry Genetics)	2018	*NR4A2* (P), *DYNC1H1* (VUS), *PNKP* (VUS), *SCN2A* (VUS)	Inconclusive
50	*GRB10* gene to confirm variant identified in lab (GeneDx	2014	*GRB10* (LP/P)	No
51	*NGLY1* evaluation for research found variant (GeneDx)	2015	*NGLY1* (LP/P)	No
52	PGL/PCC panel, evaluating for familial variant (GeneDx)	2013	*SDHD* (LP/P)	Inconclusive

“Yes” for positive diagnostic yield indicates that the patient had genetic testing results that are believed to explain the patient’s congenital cataracts based on the variant classification and inheritance pattern.

## Data Availability

Not applicable.

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
