# Peer review of "Evaluation of Genetic Testing in a Cohort of Diverse Pediatric Patients in the United States with Congenital Cataracts"

_genes, 2023, doi:10.3390/genes14030608_

Round 1
Reviewer 1 Report
This article differs from others that have reported on the results of genetic testing in children with congenital cataracts in that that 77% of the patients had systemic illnesses. I found it valuable to compare the yield with different types of testing.
Author Response
Thank you for taking the time to review our article. We greatly appreciate your time and expertise.
Reviewer 2 Report
Rossen et al. evaluated the diagnostic yield in a 20-year cohort of pediatric patients with congenital cataracts. They also reclassified the variants in accordance with ACMG guidelines. As expected WES showed the highest diagnostic yield compared to panel or microarray testing.
Major points:
All the found variants are missing in the manuscript. Are there any novel variants found? Frequencies?
In which zygosity state were the variants found?
What about the inheritance pattern of the 8% patients with a family history? Please show pedigrees.
I was not able to review section 3.6 in the current state. Please give more detailed information about the purpose of the reclassification of the 16 variants.
Minor points:
Page 5, line 151 and page 6 line 178: I do not see the purpose of referencing table 1 in this sentence.
Author Response
Major points:
Comment: All the found variants are missing in the manuscript. Are there any novel variants found? Frequencies?
Response: Thank you for this thoughtful comment. I have included a Supplementary Table 3 with specific variants and frequencies.
Comment: In which zygosity state were the variants found?
Response: Thank you for your input. We have included this information in the new Supplementary Table 3.
Comment: What about the inheritance pattern of the 8% patients with a family history? Please show pedigrees.
Response: Thank you for your comment. We have edited the family history column of Supplementary Table 2 with more information on these patients.
Comment: I was not able to review section 3.6 in the current state. Please give more detailed information about the purpose of the reclassification of the 16 variants.
Response: We appreciate your comment and have provided more detailed information about the purpose of the reclassification of the variants in Section 2.3 and 3.6.
Minor points:
Comment: Page 5, line 151 and page 6 line 178: I do not see the purpose of referencing table 1 in this sentence.
Response: Reference to Table 1 were removed from both of these locations.
Round 2
Reviewer 2 Report
Section 3.6:
I would appreciate a comment on the reason of change in classification. In particular, for the one which changed from VUS to LP. Please add a column in table S3 old/new classification.
Author Response
Thank you for your comments. We have updated Table S3 to include the variant reclassifications and have updated the manuscript to provide more information in the results and discussion section on the variant reclassification. Thank you for your efforts in strengthening this section of our paper.